# Neutralizing Antibodies Titers and Side Effects in Response to BNT162b2 Vaccine in Healthcare Workers with and without Prior SARS-CoV-2 Infection

**DOI:** 10.3390/vaccines9070742

**Published:** 2021-07-05

**Authors:** José Javier Morales-Núñez, José Francisco Muñoz-Valle, Carlos Meza-López, Lin-Fa Wang, Andrea Carolina Machado Sulbarán, Paola Carolina Torres-Hernández, Martín Bedolla-Barajas, Brenda De la O-Gómez, Paulina Balcázar-Félix, Jorge Hernández-Bello

**Affiliations:** 1Institute of Research in Biomedical Sciences, University Center of Health Sciences (CUCS), University of Guadalajara, Guadalajara 44340, Mexico; pepejavis_15@hotmail.com; 2Pediatric Service, Nuevo Hospital Civil de Guadalajara “Dr. Juan I. Menchaca”, University of Guadalajara, Guadalajara 44340, Mexico; pedmeza70@hotmail.com; 3Programmed in Emerging Infectious Diseases, Duke-NUS Medical School, Singapore 169857, Singapore; linfa.wang@duke-nus.edu.sg; 4Institute for Research on Cancer in Childhood and Adolescence, University Center of Health Sciences (CUCS), University of Guadalajara, Guadalajara 44340, Mexico; andrecaroms@gmail.com; 5Immunology Laboratory, University Center of Health Sciences (CUCS), University of Guadalajara, Guadalajara 44340, Mexico; carolina.torres.h@hotmail.com (P.C.T.-H.); paufe97@gmail.com (P.B.-F.); 6Allergy and Clinical Immunology Service, Nuevo Hospital Civil de Guadalajara “Dr. Juan I. Menchaca”, University of Guadalajara, Guadalajara 44340, Mexico; drmbedbar@gmail.com; 7University Center for Exact Sciences and Engineering (CUCEI), University of Guadalajara, Guadalajara 44340, Mexico; brenda.delao@alumnos.udg.mx

**Keywords:** COVID-19, BNT162b2, neutralizing antibodies, vaccine side effects

## Abstract

The main expected result of a vaccine against viruses is the ability to produce neutralizing antibodies. Currently, several vaccines against SARS-CoV-2 are being applied to prevent mortal complications, being Pfizer-BioNTech (BNT162b2) one of the first to be authorized in the USA and Mexico (11 December 2020). This study evaluated the efficacy of this vaccine on antibody production with neutralizing capacity and its side effects in healthcare workers with and without prior SARS-CoV-2 infection and in a group of unvaccinated individuals with prior COVID-19. The main findings are the production of 100% neutralizing antibodies in both groups after the second dose, well-tolerated adverse effects, the possible presence of immunosenescence, and finally, we support that a single dose of this vaccine in individuals with prior COVID-19 would be sufficient to achieve an immunization comparable to people without prior COVID-19 with a complete vaccination program (2 doses).

## 1. Introduction

Coronavirus disease 2019 (COVID-19) is a deadly disease caused by the SARS-CoV-2 virus and was declared a pandemic by the World Health Organization on 11 March 2020, having accumulated more than 174,000,000 global cases by June 2021. Currently, Mexico occupies fifteenth place in terms of the number of reported cases and is one of the first places in terms of mortality rate [1].

The outcome of SARS-CoV-2 infection in individuals is heterogeneous and dependent on multiple variables, mainly of comorbidities, such as diabetes and obesity, age, sex, and ethnicity. The consequences of infection can range from asymptomatic states to severe and critical illness with a high probability of death [2].

Even to this day, it is unclear whether prior SARS-CoV-2 infection protects against subsequent infection, leading to some clinical trials of the COVID-19 vaccine to exclude participants with these characteristics, although the US FDA recommends their inclusion in these trials [3].

Vaccination is the most powerful and promising public policy available nowadays to reduce infection cases, mortality and gradually lead to the cessation of the COVID-19 pandemic. Traditionally, single vaccine development takes years, even decades; however, the urgency for a COVID-19 vaccine accelerated its development, clinical trials, and approbation in a record time with the combined efforts from countries around the world [4]. Contrary to the exceptional speed of development, vaccine delivery and availability has proven to be a major challenge due to supply shortages and a limited distribution capacity in several countries [5].

One of the first vaccines to demonstrate efficacy against COVID-19 that started its global distribution was the BNT162b2 mRNA COVID-19 vaccine (Funded by BioNTech and Pfizer; ClinicalTrials.gov number, NCT04368728.), reporting a 95% protection against COVID-19 in persons 16 years of age or older after two doses [6]. BNT162b2 encodes the SARS-CoV-2 full-length spike protein, modified by two proline mutations to lock it in the prefusion conformation, mimicking the intact virus and eliciting the virus-neutralizing antibodies interaction [7].

The humoral immune response is a tool of the immune system to generate specific antibodies against pathogens, being the case of the SARS-CoV-2. Of the structural proteins produced by the SARS-CoV-2 (spike, envelope, membrane, and nucleocapsid), the spike proteins are the most antigenic, and the majority of the serological test detects antibodies against this protein [8]. It has been described that asymptomatic and mild cases of COVID-19 generated low-frequency specific IgM and IgG antibodies than those that presented severe cases [9].

The antibody concentration and affinity are generated according to the viral load and immune response from the host. Antibodies interfering in some critical functions of the antigen are those with neutralizing capacity. In SARS-CoV-2 infection, antibodies with potent binding and neutralizing activity are generally those against S protein. However, the spontaneous mutations from the virus could affect the efficacy of the neutralizing antibodies [10].

Understanding SARS-CoV-2 immunity after natural infection or vaccination also requires functional assays, such as virus-neutralizing assays. So far, virus neutralization assays (relying on cell-culture-based infection) are considered as a gold standard; however, they are assays that require appropriate biosafety facilities and are difficult to standardize [11]. Recently, a blocking ELISA (cPass™, GenScript, Piscataway, NJ, USA) has been developed and validated as a surrogate virus neutralization test (sVNT) to measure the neutralization capacity of anti-SARS-CoV-2 antibodies directed against the receptor-binding domain (RBD) of the viral spike protein (S). This test achieves 99.93% specificity and 95–100% sensitivity, with similar results to the gold standard, making it broadly accessible to a wider community for both research and clinical applications [12,13,14].

Recent studies performed in individuals from the USA have suggested that a single dose of BNT162b2 in individuals with prior SARS-CoV-2 infection could be sufficient to get a similar immunization to those individuals without prior COVID-19 with a complete vaccine scheme (two-dose) [15,16]. Therefore, we compared the presence and inhibitory capacity (measured as a percentage) of neutralizing antibodies and the side effects after immunization with the BNT162b2 vaccine in Mexican healthcare workers with and without a history of prior SARS-CoV-2 infection. Inhibitory capacity was also compared with a small group of individuals with previous COVID-19 who had not been vaccinated yet.

## 2. Materials and Methods

*Subjects and sample collection.* We included 303 healthcare workers from the Nuevo Hospital Civil de Guadalajara, Guadalajara, Jalisco, Mexico, who had been vaccinated with Pfizer-BioNTech (BNT162b2). All subjects were recruited in the *Centro Universitario de Ciencias de la Salud* (CUCS), Universidad de Guadalajara and signed an informed consent report.

Eligibility criteria included an age of 18 years or older and being a healthcare worker. Individuals undergoing treatment with immunomodulatory drugs and pregnant women were not included.

Healthcare workers were divided into three study groups: (1) individuals without prior SARS-CoV-2 infection who received Pfizer–BioNTech vaccination (n = 143), (2) individuals with prior SARS-CoV-2 infection who received Pfizer–BioNTech vaccination (n = 100), and (3) individuals with prior COVID-19 that were unvaccinated (n = 60).

All participants provided survey data on clinical and demographic characteristics if they presented prior SARS-CoV-2 infection and side effects experienced after each vaccine dose if it were the case. Among the unvaccinated individuals, there were 2 with a clinical history of severe COVID-19, 19 with a moderate course, 31 with mild illness, and 8 asymptomatic.

Individuals with prior COVID-19 had been diagnosed 3–5 months prior to the study by RT-PCR (reverse transcription-polymerase chain reaction). They were classified as follows: asymptomatic, for those without physical symptoms of COVID-19; with mild symptoms, for those who manifested fever, cough, malaise, odynophagia, headache, no dyspnea, oxygen saturation (SO2) >94%, and respiratory rate (R.R.) <20/min; with moderate symptoms, in those with SO2 > 94%, dyspnea or radiological lesions (<50% of pulmonary infiltrates), persistent fever associated with risk factors respiratory rate >20/min; and with severe COVID-19, for those with SO2 < 94% (FiO2 0.21) or R.R. > 30/min or PaO2/FiO2 < 300 or with pulmonary involvement >50%.

Individuals who denied having a history of COVID-19 (without prior COVID-19) were confirmed to be SARS-CoV-2-IgG seronegative before vaccination via IgG/IgM Rapid Test.

Peripheral blood was obtained from all individuals by venous puncture in Vacutainer tubes for serum collection. The samples for detecting IgG/IgM and neutralizing antibodies were obtained between 21–24 days after the first and second doses of vaccination. This project was conducted according to the Helsinki Declaration and approved by the Committee of Ethics and Biosecurity of the CUCS, Universidad de Guadalajara (Registry number 21-10).

*Detection of IgG/IgM and neutralizing antibodies.* The presence of IgG and IgM antibodies was determined using the kit Certum IgG/IgM Rapid Test™ cassette (Certum Diagnostics, Nuevo León, Mexico). This test is a lateral flow chromatographic immunoassay for differentiated detection of IgG (sensitivity > 99.9%, specificity 98%) and IgM (sensitivity 85%, specificity 96%) antibodies against SARS-CoV-2. This kit reacts to the presence of nucleocapsid (N) and spike (S) proteins. The protocol was performed according to the manufacturer’s instructions. First, two drops of serum were transferred to the sample well in the cassette, then two drops of buffer were added, and results were read 10 min later.

The quantification of neutralizing antibodies was performed with the cPass™ SARS-CoV-2 Neutralization Antibody Detection Kit (GenScript, Piscataway, NJ, USA), which is a blocking Enzyme-Linked Immunosorbent Assay (ELISA). The development and validation of this surrogate virus neutralization test assay have been previously reported [13,17]. This kit has also been validated for diagnosis with a 30% signal inhibition cut-off point for SARS-CoV-2 neutralizing antibody detection. The neutralization test was performed according to the manufacturer’s instructions. First, negative and positive sample controls were diluted 1:10 with the sample dilution buffer and mixed with an equal volume of HRP-conjugated RBD (60 μL and 60 μL), and then were incubated at 37 °C for 30 min. Later, 100 μL of this mixture was transferred to 96-well plates coated with recombinant hACE2 and incubated at 37 °C for 15 min. After the incubation, the supernatant was removed, and the plates were washed four times with the Wash Solution. Finally, 100 μL of tetramethylbenzidine (TMB) was added and incubated for 15 min at room temperature; the reaction was stopped with 50 μL of Stop Solution. The plates were read at 450 nm immediately after. The inhibition rate was calculated with the following formula:$$\%\;neutralization=\left(1-\frac{OD\;value\;of\;Sample}{OD\;value\;of\;Negative\;Control}\right)s\;100\%$$

*Statistical analysis.* Statistical analysis was performed using GraphPad Prism V 6.01 software. The significance level was taken at *p* < 0.05. For comparing between-group proportions, we used the chi-square (χ^2^) test and the Fisher exact test (frequencies < 5%). Data with nonparametric distribution were represented as median with interquartile range (IQR). For the analysis of variance, the Mann–Whitney U-test was applied for comparing two groups or Kruskal–Wallis for three or more, followed by Dunn’s multiple comparisons. For comparing two groups of continuous values, we used the nonparametric Wilcoxon rank-sum test. The correlations were evaluated with Spearman’s correlation. The degree of concordance between total IgG/IgM and neutralizing antibodies was determined using Cohen’s Kappa coefficient.

## 3. Results

### 3.1. Description of Study Groups

The clinical and demographic characteristics of each study group are shown in Table 1. The age and gender were similar between groups (*p* > 0.05).

In the group of individuals with prior COVID-19, we observed a greater number of individuals with comorbidities than those without prior COVID-19 (*p* = 0.0001). Of all the comorbidities, dermatitis was the most prevalent in the groups with prior COVID-19 (*p* = 0.001).

After a single vaccine dose, 5.6% of individuals in the group without prior COVID-19 were positive for IgM and 89.5% for IgG antibodies. In contrast, in the prior COVID group, 10% were positive for IgM and 100% for IgG. The difference between the percentage of positivity to IgG was statistically significant between both groups (*p* = 0.0005). After the second dose, 100% of both study groups were positive for IgG and negative for IgM. Regarding the group of unvaccinated individuals, 100% were positive for IgG antibodies and negative for IgM.

### 3.2. Vaccine-Associated Side Effects

All Table 2 shows the side effects (reactogenicity) of the BNT162b2 vaccine after the first and second dose in individuals without and with a history of prior COVID-19. After the first and second doses, myalgias, shivers, arthralgias, fever, and irritability were more prevalent in the individuals with prior COVID-19 than those without prior COVID-19 (*p* < 0.05); severe adverse events were not reported in any group.

Figure 1 shows that in the first dose, there were a greater number of vaccine-associated side effects in individuals with prior COVID-19 compared to those without prior COVID-19 (*p* < 0.0001). A similar association was observed for the second dose (*p* = 0.002).

Intra-group comparisons in individuals without prior COVID-19 showed a higher frequency of individuals with side effects in the first vaccine dose than in the second one (68% vs. 30%, *p* < 0.00001). A similar result was observed for the prior COVID-19 group since in the first dose, 90% manifested at least one side effect, while in the second dose, it was 50% (*p* < 0.00001).

The number of vaccine-associated side effects of individuals with or without prior COVID-19 was also analyzed individually at the first and second doses (Figure 2). Regarding this, no trend was observed towards an increase or decrease in the number of side effects after the first and second doses (*p* > 0.05).

The number of vaccine-associated side effects was also compared among individuals who have had mild, moderate, severe, or asymptomatic COVID-19 illness. Figure 3 shows that individuals with severe illness presented more vaccine side effects (*p* < 0.0001).

### 3.3. Generation of Neutralizing Antibodies in Response to the BNT162b2 Vaccine

The percentage of neutralization of the antibodies against SARS-CoV-2 was determined in a range of 21 to 24 days after the first and second doses of the BNT162b2 vaccine in individuals with and without previous COVID-19 (Figure 4). A statistically significant increase in the percentage of neutralization was observed in both groups after the second dose (*p* < 0.0001).

In vaccinated individuals without prior COVID-19, it was observed that 7.7% (11/143) did not have neutralizing antibodies (neutralization cut-off point <30%) after the first dose; however, 100% of them generated neutralizing antibodies 21 days after the second dose. On the other hand, 100% of the individual with prior COVID-19 had neutralizing antibodies after the first dose, increasing its neutralization percentage in the second dose. The percentage of antibody neutralization signal in individuals without prior COVID-19 was 83%, with interquartile ranges (IQR) of 62–93 after the first dose, while the median for individuals with prior COVID-19 was 98% (IQR, 97–98.4).

When analyzing the differences in the percentage of neutralization between the three study groups (Figure 5), we observed that unvaccinated individuals with prior COVID-19 had a similar neutralization percentage to those without prior COVID-19 who have received the first vaccine dose (median [IQR]: 89% (60–96) vs. 86% (69–93), respectively; *p* > 0.05). On the other hand, in both groups of vaccinated individuals, a significant increase in the percentage of neutralization after the second vaccine dose was observed (*p* < 0.001).

The percentage of neutralization signal of the antibodies generated after the second vaccine dose did not differ between individuals with or without prior COVID-19 (*p* > 0.05).

On the other hand, the neutralization percentage (median 98.3%, IQR 98.1–98.5) generated after the second vaccine dose in individuals without prior COVID-19 is similar (near to 100%) to that generated after the first dose (median 98.1%, IQR 98–98.4) in individuals with prior COVID-19 (*p* > 0.05).

### 3.4. Correlation between the Percentage of Neutralization with the Clinical and Demographic Variables

In individuals without prior COVID-19, a negative correlation (Figure 6) was observed between age and the percentage of neutralization in both the first and second doses (r = −0.23, *p* = 0.007 and r = −0.35, *p* = 0.0002, respectively). In individuals with prior COVID-19, no significant correlation was observed between this variable. Factors such as age, sex, comorbidities, vaccine side effects, and vitamin D intake were not correlated with the percentage of neutralization in any group (data not shown).

### 3.5. Concordance of the SARS-CoV-2 IgM/IgG Seropositivity with Neutralizing Antibodies Presence

Concordance of positive results to SARS-CoV-2 IgM/IgG antibodies (Certum IgG/IgM 2019-nCoV test) with the presence of neutralizing antibodies (cPass SARS-CoV-2 Neutralization Antibody Detection Kit) was compared. Results between both methods agreed between most of the positive cases (95% = 231 of 243). However, there were eight negative patients for SARS-CoV-2 IgM/IgG antibodies that tested positive for neutralizing antibodies. On the other hand, four patients showed positive to SARS-CoV-2 IgM/IgG antibodies and negative for neutralizing antibodies; these results were reproduced in duplicated. Therefore, a kappa value of 0.477 was obtained, which is interpreted as a moderate agreement between the two methods.

## 4. Discussion

Currently, different vaccines against SARS-CoV-2 have been implemented worldwide to reduce COVID-19 cases. This study focused on clarifying the different antibody responses to the BNT162b2 mRNA vaccine in individuals with and without clinical history of COVID-19; and, to compare the results of these groups with unvaccinated individuals with prior COVID-19.

The clinical characterization of the study groups confirmed the association of the presence of a more significant number of comorbidities in individuals with a history of COVID-19, as previously reported [18,19]. Interestingly, we observed a higher prevalence of dermatitis in the groups with prior COVID-19, which can be explained by two causes: in the last year, cases of occupational dermatitis have increased due to the prolonged use of personal protective equipment (PPE) and chronic stress levels [20] or as a manifestation of a drug for the treatment of COVID-19 [21].

In general, we observed that the BNT162b2 vaccine had a good tolerance in both groups, presenting mild adverse reactions that did not require additional medical attention. In addition, it is highlighted that, in both study groups, the first dose aroused more adverse reactions than the second one; this means more reactogenicity.

The vaccinated group with prior COVID-19 presented more side effects in both vaccine doses; this is consistent with other studies [6,15,22]. Interestingly, as Ebinger et al. reported, the individuals with a clinical record of severe COVID-19 had more side effects than those with asymptomatic, mild, or moderate outcomes [15].

Regarding antibody production, we made a first evaluation based on the presence of antibodies against SARS-CoV-2 through a rapid test in Cassette (Certum IgG/IgM 2019-nCoV), which detects the presence of total antibodies, evaluating both against nucleocapsid (N) and spike (S) viral proteins. After the first dose in both groups, the presence of IgM was according to that found by Wang et al., where IgG and IgM levels were maintained up to six months after vaccination and even higher than in patients with prior COVID-19 without vaccine [23]. After the first dose, the persistence of IgM could be explained by the thymus-independent response of B-cells [24].

The presence of eight patients negative to antibodies against SARS-CoV-2 through the cassette test but seropositive to neutralizing antibody test could be a false negative result of the cassette test, as it contains nonspecific proteins that could not bind to the antibodies of patients [25]. On the other hand, the four positive cases for antibodies against SARS-CoV-2 using the cassette test with negative detection of neutralizing antibodies can be explained because this second method specifically detects the RBD portion of S protein. Therefore, the antibodies of these four patients could not neutralize the RBD portion, but it does not exclude that they exert other mechanisms of antibodies defense [26,27,28].

Since the BNT162b2 vaccine releases mRNA encoding only protein S, the expected elicited response is the specific production of IgG antibodies (S-RBD) and not others such as IgG (N) antibodies [29].

Regarding neutralizing antibodies, we observed a significant increase in the percentage of neutralization after the second BNT162b2 dose in both study groups. This finding agrees with that reported by others [29,30]. Interestingly, our study shows that 11 patients did not present neutralizing antibodies in the first vaccine dose, which differs from other studies where all patients develop neutralizing antibodies at the first dose [15,31]. It should be noted that this occurred in the group of people vaccinated without prior COVID-19, who generated neutralizing antibodies after the second dose.

The above does not necessarily mean that the vaccine did not induce the production of antibodies against SARS-CoV-2 in the first dose, but rather that a dose for these patients did not reach the threshold to produce antibodies with neutralizing capacity.

For the vaccinated individuals with prior COVID-19, the first dose was sufficient to induce the production of neutralizing antibodies in all of them, maintaining and increasing the percentage of neutralization with the second dose, consistent with other descriptions [32].

An important consideration is that the mean of the neutralization percentage after the second vaccine dose reached similar levels in individuals with and without prior COVID-19, similar to that found in a recent report [15]. However, two people reached a low percentage of neutralizing antibodies (33% and 45%, respectively) at the second dose in the group without prior COVID-19. Although two cases are too few to establish a conclusion, this observation could be explained by two factors, the first by the immunosenescence phenomenon, being two persons over 55 years of age, and the second factor for having not reported reactogenicity to vaccination, as Sprent et al. suggested, the presence of side effects is associated with the activation of the immune response [33].

Regarding the group with prior COVID-19, we observed a person with a neutralizing antibodies value of 60% after the second dose; this could be explained because that person had a history of severe COVID-19 and a prolonged high-dose glucocorticoid therapy. Thus, a residual immunosuppressive effect of this treatment could explain the low neutralizing percentage [34].

The phenomenon of immunosenescence [35,36] was observed in the vaccinated group without prior COVID-19. This finding was reflected by a negative correlation between the percentages of neutralizing capacity with age, indicating that young people have a better response to antibody production with neutralizing capacity than those of an older age. Therefore, to our knowledge, this study is a pioneer in reporting this association for the BNT162b2 vaccine and is agree with those observed by Walsh et al., who reported lower antigen-binding IgG and virus-neutralizing responses in participants 65 to 85 years of age than in those 18 to 55 years of age [7]. In the group of those vaccinated with prior COVID-19, this trend was not observed since they possibly had a previous immunogenic encounter that favors the reduction in the age gap in the generation of neutralizing antibodies.

The generation of neutralizing antibodies in unvaccinated individuals with prior COVID-19 has been poorly explored. In these individuals, we observed similar levels of neutralizing antibodies as the group without prior COVID-19 who received the first dose of vaccine. Interestingly, the neutralization percentage generated after the first vaccine dose in individuals with prior COVID-19 is similar (near to 100%) to that generated after the second dose in individuals without prior COVID-19. This result suggests that a single dose of the BNT162b2 vaccine could be sufficient to confer a similar immunization in those patients with a previous history of COVID-19 (individuals vaccinated at least 3–5 months after SARS-CoV-2 infection). This hypothesis has been recently supported by other authors [15,16,22]. However, this study does lack information on how long these antibody titers could be maintained in individuals who received the first dose after prior infection and those with a second dose and no prior infection. This information would have further strengthened the proposal for single-dose vaccination for individuals with prior SARS-CoV-2 infection. Moreover, the ability to clinically correlate neutralizing activity with a degree of immunity or protection from reinfection is still pending and will require further investigation.

A weakness of our study is that we did not determine neutralizing antibody concentrations before vaccination. These would have helped to compare convalescent antibody levels with vaccine-induced levels. Another weakness could be the use of a blocking ELISA for the measurement of neutralizing antibodies; however, this test has demonstrated similar results to the standard gold (VNT) [12,14,37]. Based on the comparison of the results obtained by our methodology and those obtained by VNT from other studies [29,30], we did not observe any difference. Thus, we consider that the results obtained with our methodology could be considered for clinical extrapolations.

A further consideration is that we evaluated the presence of neutralizing antibodies only up to 24 days after the second dose; therefore, it is necessary to analyze this vaccine’s effect regarding its ability to produce neutralizing antibodies months after its application. Khoury et al. implemented a predictive mathematical model, concluding that the capacity of various vaccines to maintain the production of neutralizing antibodies will drop exponentially after 250 days post-vaccine [38], and Chia et al. demonstrate this in the natural infection [17]. As a limitation, our study does not currently have results of the follow-up of neutralizing antibody titers of both study groups. Further investigations must be monitory at least 6, 12, and 18 months after vaccination for the aforementioned.

## 5. Conclusions

In conclusion, our results showed different neutralizing antibody generation dynamics in response to the BNT162b2 mRNA vaccine in individuals with and without a history of SARS-CoV-2 infection. The findings support the proposal that a single dose of this vaccine would be sufficient to achieve an immunization comparable to people without prior COVID-19 with a complete vaccination program. The previous is very relevant for decision-making in vaccination schemes in countries with low access to COVID-19 vaccines, such as Mexico.

## Figures and Tables

**Figure 1 vaccines-09-00742-f001:**
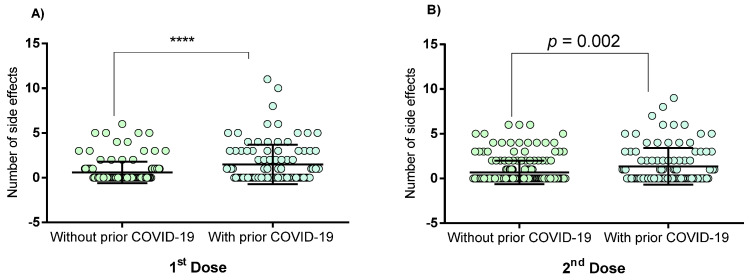
Side effects to Pfizer–BioNTech vaccine in patients with and without prior COVID-19. (**A**) Side effects to a first dose; (**B**) Side effects to the second dose. Data were calculated using the Mann–Whitney U-test. Values are provided as medians and interquartile ranges. **** *p* < 0.0001.

**Figure 2 vaccines-09-00742-f002:**
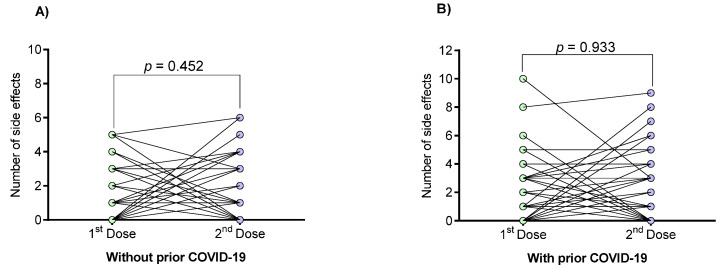
Numbers of vaccine-associated side effects in individuals with and without prior COVID-19 after first and second doses. (**A**) Side effects in individuals without prior COVID-19; (**B**) Side effects in individuals with prior COVID-19. Data were calculated using the Mann–Whitney U-test. Values are provided as medians and interquartile ranges. Differences were calculated by the Wilcoxon signed-rank test.

**Figure 3 vaccines-09-00742-f003:**
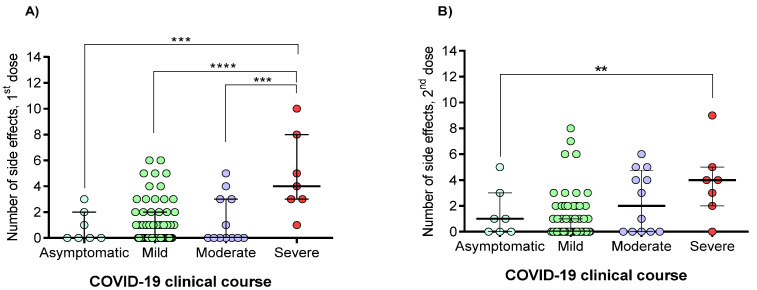
Relationship between prior COVID-19 clinical course with the number of vaccine-associated side effects. (**A**) Side effects after a singles dose; (**B**) Side effects after the second dose. The difference between all groups was calculated with the Kruskal–Wallis test, followed by Dunn’s multiple comparison test. Data are provided as median and interquartile ranges. **, *p* < 0.01; ***, *p* < 0.001; ****, *p* < 0.0001.

**Figure 4 vaccines-09-00742-f004:**
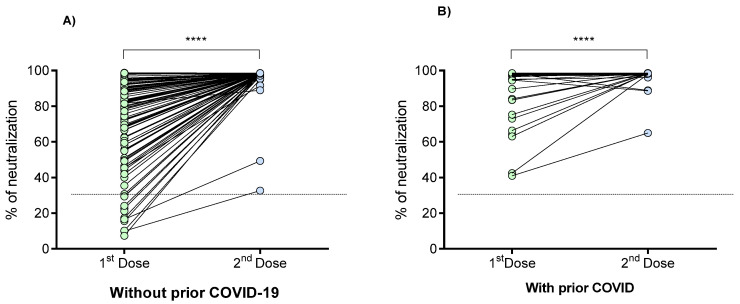
Percentage of neutralizing signal of antibodies generated in response to the BNT162b2 vaccine in individuals with and without prior COVID-19. (**A**) Individuals without prior COVID-19; (**B**) Individuals with prior COVID-19. Differences were calculated by the Wilcoxon signed-rank test. The dotted line indicates the cut-off point for the neutralization test (>30%). ****, *p* < 0.0001.

**Figure 5 vaccines-09-00742-f005:**
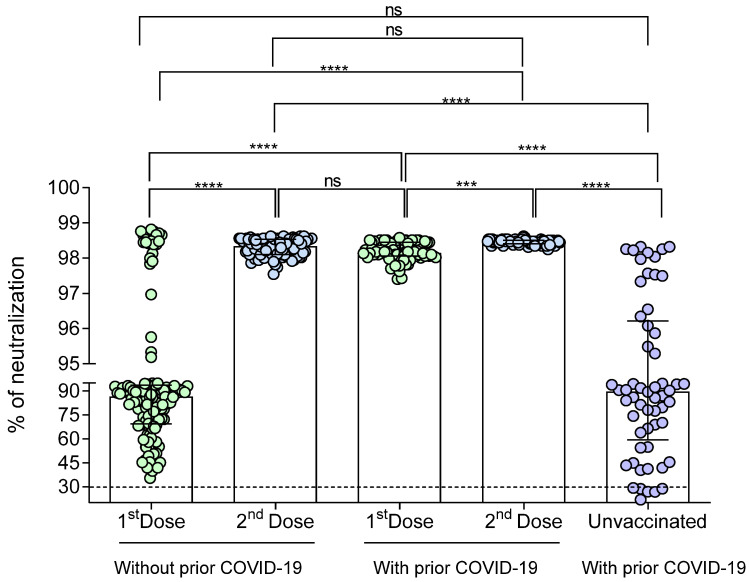
Comparison of the neutralization percentages of the antibodies generated in response to the BNT162b2 vaccine or single SARS-CoV-2 infection. The difference between all groups was calculated with the Kruskal–Wallis test, followed by Dunn’s multiple comparison test. The data are provided as medians and interquartile ranges. ***, *p* < 0.001; ****, *p* < 0.0001; ns, not significant (*p* > 0.05). Outliers were removed for data analysis.

**Figure 6 vaccines-09-00742-f006:**
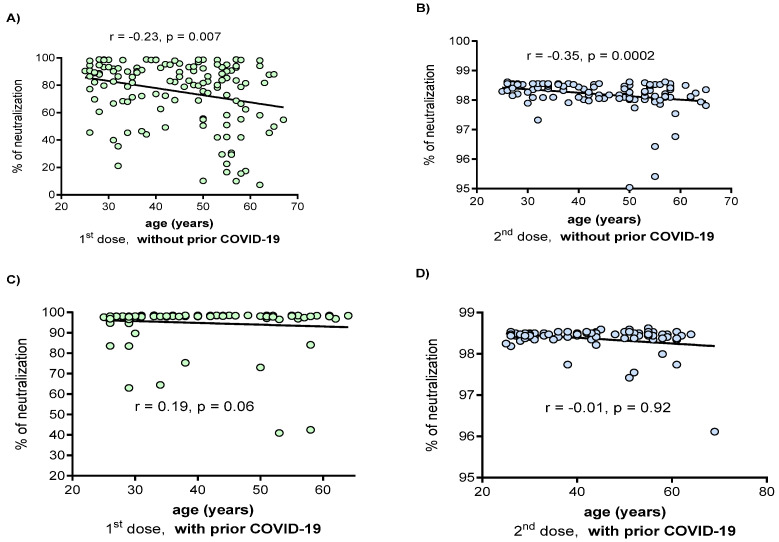
Correlation between the age and neutralization percentage in vaccinated individuals with and without prior COVID-19. (**A**) the first dose in patients without prior COVID-19; (**B**) the second dose in patients without prior COVID-19; (**C**) the first dose in patients with prior COVID-19; (**D**) the second dose in patients with prior COVID-19. Correlations were evaluated by the Spearman test. Outliers were removed for data analysis.

**Table 1 vaccines-09-00742-t001:** Clinical and demographic characteristics of the study groups.

	Immunized with BNT162b2 Vaccine			*p*-Value
	Yes		No	
	without Prior COVID-19 n = 143	with Prior COVID-19 n = 100	with Prior COVID-19 n = 60	
Age (years), mean ± SD	45 ± 12	41 ± 12	42 ± 11	0.06
Sex, n (%)				
Male	58 (41)	42 (42)	24 (40)	0.962
Female	85 (59)	58 (58)	36 (60)	
Comorbidity, n (%)				
At least one comorbidity	62 (43)	70 (70)	31 (52)	0.0001
Diabetes	2 (1.4)	2 (2)	2 (3.3)	0.66
SAH	8 (6)	9 (9)	4 (6.6)	0.581
Allergic diseases	15 (10)	20 (20)	6 (10)	0.089
Dermatitis	2 (1.4)	12 (12)	2 (3.3)	0.001
Overweight	35 (24)	27 (27)	17 (28)	0.793
Positivity to antibodies, n (%)				
1st dose				
IgM	8 (5.6)	10 (10)		0.297
IgG	128 (89.5)	100 (100)		0.0005
2nd dose, n (%)				
IgM	0 (0)	0 (0)	-	-
IgG	100 (100)	100 (100)	-	1.00
Vitamin D intake (4000 IU/day), n (%)	42 (29)	32 (32)	20 (33)	0.812

SD, standard deviation. SAH, Systemic arterial hypertension. *p*-values were calculated by Fisher’s exact or ANOVA test.

**Table 2 vaccines-09-00742-t002:** Side effects to BNT162b2 vaccine after the first and second dose.

Side Effects to the Vaccine	Dose	Immunized with BNT162b2 Vaccine		*p*-Value
		Without Prior COVID-19 N = 143 n (%)	With prIor COVID-19 N = 100 n (%)	
None	1st	46 (32)	10 (10)	<0.0001
	2nd	100 (70)	50 (50)	0.002
At least 1 side effect	1st	97 (68)	90 (90)	<0.0001
	2nd	43 (30)	50 (50)	0.002
Myalgia	1st	16 (11)	27 (27)	0.002
	2nd	15 (10.5)	28 (28)	0.0006
Odynophagia	1st	5 (3.5)	2 (2)	0.703
	2nd	3 (2.1)	4 (4)	0.450
Cough	1st	4 (2.8)	4 (4)	0.720
	2nd	3 (2.1)	1 (1)	0.645
Shivers	1st	8 (5.56)	14 (14)	0.038
	2nd	11 (7.7)	17 (17)	0.039
Rhinorrhea	1st	4 (2.8)	7 (7)	0.207
	2nd	6 (4.2)	7 (7)	0.391
Headache	1st	34 (24)	26 (26)	0.692
	2nd	22 (15.4)	24 (24)	0.091
Arthralgias	1st	4 (2.8)	20 (20)	<0.0001
	2nd	9 (6.3)	14 (14)	0.048
Fever	1st	7 (4.8)	13 (13)	0.031
	2nd	5 (3.5)	16 (16)	0.0009
Dysgeusia	1st	2 (1.4)	4 (4)	0.232
	2nd	2 (1.4)	1 (1)	1.00
Irritability	1st	4 (2.8)	10 (10)	0.024
	2nd	4 (2.8)	10 (10)	0.024
Chest pain	1st	3 (2.1)	4 (4)	0.450
	2nd	2 (2.8)	1 (1)	1.00
Diarrhea	1st	3 (2.1)	4 (4)	0.450
	2nd	7 (4.8)	4 (4)	0.207

*p*-value was calculated with Fisher’s exact.

## Data Availability

The data that support the findings of this study are available on request from the corresponding author.

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
