# Peer review of "Neutralizing Antibodies Titers and Side Effects in Response to BNT162b2 Vaccine in Healthcare Workers with and without Prior SARS-CoV-2 Infection"

_vaccines, 2021, doi:10.3390/vaccines9070742_

Round 1

Reviewer 1 Report

The  authors evaluate the efficacy of Pfizer vaccine in health care workers with or without prior SARS-CoV2 infection. Authors find 100% antibody response in both groups after second dose. Interestingly,individuals with prior covid infection elicited neutralizing antibodies after single dose of the vaccine, that were comparable to levels generated after second dose in individuals with no history of infection.

This is a well written paper, data clearly presented and the findings are of interest.

However, the study does lack information of how well  and how long these antibody titres are maintained in individuals who received first dose after prior infection and those with second dose and no prior infection.  This information would have further strengthened the proposal for single dose vaccination for individuals with prior covid infection.

Again what are the chances of reinfection in these two cases. For the cohort studied, were any reinfections reported?

I suggest the authors do include these drawbacks/limitation of the study in the discussion.

Author Response

Reviewer 1

Point 1.  This is a well written paper, data clearly presented and the findings are of interest. However, the study does lack information of how well and how long these antibody titres are maintained in individuals who received first dose after prior infection and those with second dose and no prior infection. This information would have further strengthened the proposal for single dose vaccination for individuals with prior covid infection. Again what are the chances of reinfection in these two cases. For the cohort studied, were any reinfections reported? I suggest the authors do include these drawbacks/limitation of the study in the discussion.

Answer: Thank you for your valuable comments. We added in the discussion a statement about this limitation (highlighted in green color). For reasons of research ethics, we cannot follow the antibody dynamics of patients with previous COVID-19 with a single dose of vaccine. However, this manuscript can give the guideline to do it later. We are currently following the patients included in this study, but we will carry out determinations at 6, 12, and 18 months. We will report these findings as soon as the required time passes. So far (3 months from the second dose), we have not received notification of a positive case for COVID-19 after vaccination. Now, the manuscript was checked by a native English-speaking colleague.

Reviewer 2 Report

The manuscript by Morales-Nuñez et al. describes their research on the neutralizing antibodies titers and side effects in response to BNT162b2 vaccine in healthcare workers with and without prior SARS-CoV-2 infection. The main findings are the production of 100% neutralizing antibodies in both groups after the second dose, well-tolerated adverse effects, the possible presence of immunosenescence. However, I have only one question for this manuscript. In Figure 4, author mention that in vaccinated individuals without prior COVID-19, it was observed that 7.7% (11/143) did not have neutralizing antibodies after the first dose. But, two of them generated lower neutralizing antibodies 21 days after the second dose. Why? The author must elaborate and discuss that.

Author Response

Reviewer 2

Point 1.  The manuscript by Morales-Nuñez et al. describes their research on the neutralizing antibodies titers and side effects in response to BNT162b2 vaccine in healthcare workers with and without prior SARS-CoV-2 infection. The main findings are the production of 100% neutralizing antibodies in both groups after the second dose, well-tolerated adverse effects, the possible presence of immunosenescence. However, I have only one question for this manuscript. In Figure 4, author mention that in vaccinated individuals without prior COVID-19, it was observed that 7.7% (11/143) did not have neutralizing antibodies after the first dose. But, two of them generated lower neutralizing antibodies 21 days after the second dose. Why? The author must elaborate and discuss that

Answer: Thank you for your valuable comments. We elaborate and discuss that point out by the reviewer. We added the following paragraphs to the discussion section (highlighted in yellow): However, two people reached a low percentage of neutralizing antibodies (33% and 45%, respectively) at the second dose in the group without prior COVID-19. Although both two cases are few to establish a conclusion, this observation could be explained by two factors, the first by the immunosenescence phenomenon, being two persons over 55 years of age, and the second factor for having not reported reactogenicity to vaccination, as Sprent et al., suggested that the presence of side effects is associated with the activation of the immune response [35].

Regarding the group with prior COVID-19, we observed a person with a neutralizing antibodies value of 60% after the second dose; this could be explained because that person had a history of severe COVID-19 and a prolonged high-dose glucocorticoid therapy, thus, a residual immunosuppressive effect of this treatment could explain that low neutralizing percentage [36].